# Intraoral Scanners for In Vivo 3D Imaging of the Gingiva and the Alveolar Process

**DOI:** 10.3390/jcm11216389

**Published:** 2022-10-28

**Authors:** Jonas Winkler, Anton Sculean, Nikolaos Gkantidis

**Affiliations:** 1Department of Orthodontics and Dentofacial Orthopedics, School of Dental Medicine, University of Bern, CH-3010 Bern, Switzerland; 2Department of Periodontology, School of Dental Medicine, University of Bern, CH-3010 Bern, Switzerland

**Keywords:** gingiva, mucosa, alveolar process, digital dental model, three-dimensional imaging, surface model

## Abstract

This study aimed to assess the reliability of two intraoral surface scanners for the representation of the alveolar process in vivo. Complete maxillary scans (CS 3600, Carestream and TRIOS 3, 3Shape) were repeatedly obtained from 13 fully dentate individuals. Scanner precision and agreement were tested using 3D surface superimpositions on the following reference areas: the buccal front teeth area, the entire dental arch, the entire alveolar process, or single teeth by applying an iterative closest point algorithm. Following each superimposition, the mean absolute distance (MAD) between predefined 3D model surfaces was calculated. Outcomes were analyzed through non-parametric statistics and the visualization of color-coded distance maps. When superimpositions were performed on the alveolar process, the median scanner precision was below 0.05 mm, with statistically significant but negligible differences between scanners. The agreement between the scanners was approximately 0.06 mm. When single-tooth superimpositions were used to assess the precision of adjacent alveolar soft-tissue surfaces, the median error was 0.028 mm, and there was higher agreement between the scanners. The in vivo reliability of the intraoral scanners in the alveolar surface area was high overall. Single-tooth superimpositions should be preferred for the optimal assessment of neighboring alveolar surface areas relative to the dentition.

## 1. Introduction

The proper assessment and monitoring of the alveolar processes over time is imperative for all dental disciplines that aim to preserve the natural teeth in the dentition [1], to improve tooth position [2], or to treat any pathologies and restore missing teeth [3,4]. Consequently, there are several methods to assess the alveolar and gingival structures and monitor possible changes over time. These include direct measurements, e.g., through a probe or a caliper [3,4], measurements on physical or digital dental models [5,6], or radiographic assessments [7,8,9]. Each method has its case-specific pros and cons and should be used if certain indications are present. For example, it is crucial to avoid subjecting patients to unnecessary risks, such as radiation exposure, or to extra costs that might have been avoided. However, since the alveolar process is comprised of bone and soft tissues, a fusion of 3D radiographs and surface scans is needed to detect changes in each tissue [10].

Intraoral dental scans, which are rapidly being incorporated into everyday clinical practice, offer a convenient, risk-free, and relatively low-cost 3D imaging tool that offers detailed 3D information on the surface morphology of dental, palatal, and alveolar structures [6,11]. An intraoral scan has several applications in contemporary dentistry, including the monitoring of dental tissues or restorations [12,13,14], the assessment of tooth positional changes [5,15], and the construction of dental restorations [16]. It has been suggested that every dental patient should have an intraoral scan at the early stages of permanent dentition to be used as the reference for comparisons with future scans [12,14]. This way, changes in oral structures can be monitored with high accuracy, and developing problems can be detected in the early stages, enabling in many cases the avoidance of further complications and the high costs required for the management of progressed complex problems.

So far, various studies have tested the reliability of intraoral scanners in vivo, but they primarily regarded the dental [11,17] and palatal structures [6]. There are a few in vivo reports [18,19] that assessed the gingival surface, but they either included only a small area extending approximately 1 mm below the free gingival margin [18] or were focused on the esthetic zone [19] or on specific teeth [20]. There are also other reports that were focused on the free gingival margin limits [21]. A thorough assessment of the reliability of the entire alveolar process surface acquired in vivo through intraoral 3D scanning in individuals with full dentition would offer valuable information.

Therefore, the aim of this study was to assess the precision of two widely used intraoral surface scanners, as well as their agreement, on the representation of the original morphology of the entire alveolar process in vivo. For this, complete maxillary 3D models were obtained under actual clinical conditions, superimpositions of the subsequent 3D models were performed on various anatomical structures, and the implications of each approach for outcome interpretation are discussed.

## 2. Materials and Methods

### 2.1. Sample

The material for this study was obtained in the context of a larger project and was already used and reported in two publications regarding the assessment of the accuracy of the whole dental arch and the palatal region [6,11]. However, the information necessary for the readers to comprehensively assess the present study is repeated here. Further details can be found in the previous reports [6,11]. For the purposes of the present study, data from 13 adults (9 males and 4 females; 27–52 years old), comprising all subjects recruited according to the research protocol, were used.

These subjects fulfilled the following eligibility criteria:-No extremes of palatal shape and no visible edema in the attached or removable buccal or on the palatal mucosa (visual inspection by two investigators, with disagreements resolved by consensus).-No extreme malocclusion patterns, no crossbite, and no large asymmetries (visual inspection by two investigators, with disagreements resolved by consensus).-Participants of European (white) ancestry. This criterion was applied for consistency since this group was over-represented in the place of sample selection.-Before or more than 2 years after the end of any previous orthodontic treatment.

During the recruitment phase, one subject was excluded based on the last reported criterion, and all other subjects were included without any disagreements that needed to be resolved.

In the absence of any existing data for the buccal and palatal sides of the alveolar processes, an a priori power calculation for the tested outcomes was not possible. Thus, we decided to use all available data, which previously provided satisfactory power for analogous outcomes [6,11,22].

### 2.2. Data Acquisition

The data acquisition process was thoroughly described previously [11] in a study with a different aim. The relevant information is repeated here.

Intraoral digital 3D models were obtained using the CS 3600 (Carestream, Atlanta, GA, USA, Software CS Imaging Version 7.0.23.0.d2) and the TRIOS 3 (3Shape, Copenhagen, Denmark, Software Version 1.4.7.5) scanners, with the following sequence:3D scans with CS 3600 (two times)3D scans with TRIOS 3 (two times)3D scan with CS 3600 (one time)

All scans were performed by an author who had more than two years of experience with regular clinical use of the tested intraoral scanners. The same investigator, who had adequate experience in the field [6,11] and was supervised by the senior author, performed all steps of data generation.

As reported previously [6,11], all participants were in a horizontal position on the dental chair during image acquisition. The five intraoral scans were obtained consecutively in the order described above after proper tooth drying. The scanning of the maxilla started with the second molar in the first quadrant and ended at the second molar in the second quadrant. The scanning of palatal soft tissues started at the palatal side of the central incisors and moved distally back to the level of the distal end of the second molars. Missed areas were rescanned before completing each scanning session. Unbroken and smooth digital images were considered eligible for inclusion in the study. The entire maxillary arch, including the palatal surface, was scanned in about 2–3 min using the CS3600 scanner and in about 2 min using the TRIOS 3 scanner.

The 3D surface models from all intraoral scanners were exported as STL files through each scanner’s software (CS 3600: CS Imaging, Version 7.0.23.0.d2; TRIOS 3: Trios, Version 1.4.7.5). These STL files were then imported into Viewbox 4 software (Version 4.1.0.1 BETA 64, dHAL Software, Kifissia, Greece).

### 2.3. Assessment of Precision and Agreement between Scanners

All 3D surface models were manually cropped within 7 mm apical to the gingival margin area and distal to the first molar (Figure 1a). This soft-tissue area is the most relevant for the support of teeth or dental restorations and is expected to include the total height of keratinized gingiva [23], which are fixed to the underlying bone. The surface of the movable mucosa is not morphologically stable, and thus the proper scanning of this area with the current intraoral scanners is neither realistic nor accurate. Following this process, the subsequent 3D models were exported and saved again as STL files that comprised the final 3D models analyzed in the study. Each of these models consisted of approximately 80,000 and 120,000 vertices for CS 3600- and TRIOS 3-derived models, respectively.

To test the precision and the agreement of the scanners on overall alveolar surface imaging, the buccal and palatal gingival and oral mucosa surfaces, from the gingival margins of all teeth (16–26) to 7 mm apically, were used as superimposition references (Figure 1b).

For the testing of the alveolar surface relative to dental structures, a previously verified accurate area [11] at the buccal surfaces of the maxillary anterior teeth was used as a superimposition reference (Figure 1d). Alternatively, the entire clinical crowns of all teeth between the first maxillary molars were also used for the same purpose (Figure 1c).

The agreement and precision of the two scanners were also tested on a small spatial scale by superimposing the repeatedly acquired models on a single tooth each time (maxillary right central incisor, maxillary right canine, and maxillary left second premolar; Figure 1e) and measuring the corresponding distances of the adjacent alveolar structures, as described below. Six measurement areas were selected for this purpose on the 3D models (two for each tooth). On the buccal side, the first region (area B11) was a circular area of approximately 2 mm diameter (100 vertices), centered 2 mm apical to the deepest point of the gingival margin area of the maxillary right central incisor. The second region (area B13) was analogous but on the maxillary right canine, and the third region (area B25) was on the maxillary left second premolar. Corresponding areas on the palatal side were also assessed (P11, P13, and P25, respectively) (Appendix A).

The agreement and precision of the intraoral scanners (TRIOS 3 and CS 3600) were tested after superimposing the maxillary 3D models on one of the different superimposition reference areas described above (buccal front teeth area, entire dental arch, entire alveolar process, and single teeth). The final 3D models were superimposed using Viewbox 4 software to apply an iterative closest point algorithm (ICP) [24] with the following settings: 100% estimated overlap of meshes, matching point to plane, exact nearest neighbor search, 100% point sampling, exclude overhangs, and 50 iterations. The algorithm was repeatedly applied until the minimum distance between the matched models was obtained (usually 4–5 times).

Following each superimposition, the testing variable was the mean absolute distance (MAD) between the corresponding 3D model surfaces of the entire buccal and palatal alveolar process area or the 2 mm circular regions apical to the gingival margins of specific teeth, as reported above (Appendix A). Respective color-coded distance maps were generated, showing the buccal front, buccal right, buccal left, and palatal sides.

### 2.4. Statistical Analysis

The statistical analysis was carried out using SPSS software (IBM SPSS Statistics for Windows, Version 27, Armonk, NY, USA: IBM Corp.), following a similar approach to a previous analogous study [11]. Raw data were tested for normality through the Shapiro–Wilk test and did not have a normal distribution in certain cases. Thus, non-parametric statistics were applied. The differences in the measured variables were tested in a paired manner through the Friedman test. In case of significant outcomes, pairwise comparisons were performed through the Wilcoxon signed-rank test. In all cases, a two-sided significance test was carried out at an alpha level of 0.05. The Bonferroni correction was applied for pairwise a posteriori multiple comparison tests.

## 3. Results

When superimpositions were performed on the alveolar process, the median precision of the two scanners was below 0.05 mm, with significant differences between them (Friedman test, *p* = 0.006; TRIOS 3 median: 0.039 mm, range: 0.025–0.127 mm; CS 3600 median: 0.049 mm, range: 0.028–0.131 mm) (Figure 2). However, the magnitude of this difference was approximately 10 µm, and it was considered negligible. Furthermore, pairwise comparisons did not identify any significant differences between any pairs of repeated scans (*p* > 0.01, Appendix A). For the same outcome, the agreement between scanners was at the level of 0.06 mm (median: 0.057 mm, range: 0.034–0.091 mm), with all differences remaining below 0.1 mm (Figure 3). A visual inspection of the relevant color-coded distance maps revealed a tendency for increased imprecision towards the direction of the removable mucosa, which primarily concerned the areas of frenum insertion. Higher imprecision was also evident in the buccal compared to the palatal areas, primarily attributed to the latter observation. No clear difference was observed between the anterior and posterior areas of the arch. In a few cases, higher imprecision was observed in specific areas of certain pairs of scans, considering both used scanners, which exceeded 0.2 mm. No specific pattern regarding the location of differences was evident in these cases (Figure 4). All of the aforementioned findings were also observed considering the agreement between the two scanners (Figure 5).

When superimpositions were performed on the buccal surfaces on the front teeth, the median scanner precision on the alveolar process was approximately 0.07 mm, with no significant differences between the scanners (Friedman test, p = 0.109; TRIOS 3 median: 0.062 mm, range: 0.039–0.145 mm; CS 3600 median: 0.077 mm, range: 0.040–0.709 mm) (Figure 2). For this outcome, the agreement between scanners was at the level of 0.09 mm (median: 0.095 mm, range: 0.071–0.148 mm), with all differences remaining below 0.15 mm (Figure 3). The associated color-coded distance maps revealed a tendency for increased imprecision towards the direction of the removable mucosa, as described above for the superimposition on the alveolar process. On the contrary, when we superimposed on the front teeth, higher imprecision was evident in the palatal compared to the buccal areas. Additionally, with this superimposition, slightly higher imprecision was detected in the posterior compared to anterior alveolar areas. Finally, a larger number of cases showed higher imprecision, exceeding 0.2 mm, in specific areas of certain pairs of scans, considering both scanners. No specific pattern regarding the location of the differences was evident in these cases (Appendix A). Only the latter two findings were evident when considering the agreement between the two scanners (Figure 5).

When superimpositions were performed on the entire dentition, the scanner precision in the region of the alveolar process was approximately 0.05 mm, with a statistically significant but quite small difference of 10 µm between the two scanners (Friedman test, p = 0.016; TRIOS 3 median: 0.042 mm, range: 0.029–0.137 mm; CS 3600 median: 0.055 mm, range: 0.030–0.439 mm) (Figure 2). Furthermore, pairwise comparisons did not identify any significant differences in the precision between any pairs of repeated scans (*p* > 0.01) (Appendix A). For the same outcomes, the agreement of the scanners was at the level of 0.06 mm (median: 0.063 mm, range: 0.048–0.110 mm), with almost all differences remaining below 0.1 mm (Figure 3). For this superimposition, a visual inspection of the color-coded distance maps of individual cases revealed similar patterns as those observed following the superimposition of repeated scans on the alveolar process, with no detectable differences between the buccal and palatal areas (Figure 5 and Appendix A).

Regarding the precision of the scanner in specific alveolar areas relative to corresponding teeth (#11, #13, and #25) (Appendix A), there were no differences between scanners (Friedman test: *p* > 0.05), apart from a small difference regarding the buccal side of the maxillary right incisor (*p* = 0.048). However, even in the latter case, no significant differences were detected between any compared pair of repeated scans (Wilcoxon signed-rank test: *p* > 0.01, Appendix A). The median difference of any pair of repeated scans was consistently below 0.05 mm, whereas the maximum detected difference was 0.165 mm. Furthermore, there was a tendency for higher imprecision towards posterior areas of the arch in a few specific cases, but overall similar precision was observed in all tested areas, which varied close to 0.025 mm (overall median: 0.028 mm, Figure 6). For this outcome, similar findings were evident considering the agreement between scanners (overall median: 0.044 mm, maximum difference: 0.153 mm, Figure 3).

## 4. Discussion

The accurate assessment of the alveolar process is important for various fields of dentistry that may affect the relevant structures through interventions or from diagnostic, progress evaluation, or appliance/restoration construction perspectives [25]. The present study performed a thorough in vivo assessment of the precision and the agreement of two intraoral surface scanners on the soft-tissue surfaces of the alveolar process in fully dentate individuals. When considering the precision of only the alveolar soft-tissue surfaces, irrespective of other intraoral structures, such as the teeth, the scanners showed similar overall precision, with a difference between repeated scans below 50 μm. Similar outcomes were evident for the agreement between the scanners. Therefore, this reliability level can be considered high for clinical outcomes. In a few cases though, higher imprecision and disagreement were observed, which exceeded 200 μm. Apart from a tendency towards higher imprecision and reduced agreement when moving from the attached gingiva to the removable mucosa, there was no other clear spatial pattern within the entire tested surface.

When similar assessments were performed after the superimposition of repeated models on the entire dentition, there was a very slight reduction in the overall precision of the alveolar surface assessments, though in limited cases, particularly of the CS3600 scanner, there were large errors. Higher imprecision was detected when superimpositions were performed on the buccal surfaces of the front teeth, with the difference between repeated scans in most cases still remaining below 100 μm. Similar levels of agreement were evident between the two scanners. However, imprecision and differences between scanners were higher when individual cases were considered. On the contrary, when single-tooth superimpositions were used to assess the precision of adjacent alveolar soft-tissue surfaces, the overall error was 28 μm and never exceeded 200 μm. The agreement between the scanners was also maximized with the latter approach. Based on the above considerations, we suggest single-tooth superimpositions as the method of choice when adjacent alveolar soft-tissue structures should be evaluated with high accuracy. This will also facilitate the standardization of measurement methods and thus the comparability of outcomes obtained from different studies [26].

To elaborate on this important aspect, when superimpositions were performed on the buccal surfaces on the front teeth, the two scanners showed similar precision in the region of the alveolar process that was, however, lower than that detected when superimpositions were performed on the alveolar process itself. Furthermore, in contrast to the latter superimposition, both scanners showed higher imprecision in the palatal compared to the buccal areas, slightly higher imprecision in the posterior compared to anterior areas, and a larger number of cases with higher imprecision in specific areas of certain scans. These inconsistencies were primarily attributed to the selected superimposition reference area. When the buccal anterior tooth surfaces were used to superimpose the corresponding models, the best fit registration provided a local maximum congruence at the reference area, but the further an assessment area was from the reference area, the higher the error. This has already been reported in various previous studies [6,11,27,28] and highlights the need for proper outcome interpretation relative to the reference area used, which should be defined according to the outcome of interest. Furthermore, these findings demonstrate that the imprecision was smaller when local changes in a certain area of the alveolar process were assessed compared to changes in the alveolar process relevant to the teeth. Again, as reported previously [6,11,12,13,14], when larger areas were considered, the scanner imprecision was higher than that expected in smaller areas. This was attributed to the individual image stitching during acquisition and processing, which was required to obtain the complete 3D model but might cause small inaccuracies at each step that have an additive effect when larger areas are considered. The findings regarding the agreement between scanners support these arguments, although in certain cases they followed slightly different patterns. This might be attributed to differences in the hardware and software used by each scanner.

Our literature search identified a few previous in vivo studies that tested the alveolar surface area [18,19]. One study included a gingival margin area of approximately 1 mm vertical height, along with the dental arch [29], but performed an overall assessment of the entire surface model without differentiating between the gingival and tooth structures. Another in vivo study [19] assessed a larger alveolar soft-tissue area, but it was limited to the upper anterior buccal soft tissues. In the latter study, the superimposition of corresponding surface models was performed only at the anterior teeth. Thus, the precision of solely the alveolar soft-tissue surfaces was not evaluated. This approach offered an overall assessment, averaging the errors of teeth and soft tissues, which could confound the alveolar surface outcomes. Furthermore, in the aforementioned study the intraoral scanner models were compared to models derived from conventional impressions. However, the impression material might exert pressure in the soft tissues under study [30], altering the actual surface that is used as a reference for accuracy testing. This, together with the superimposition approach described above, might have skewed the outcomes. Indeed, the precision values reported by that study regarding the alveolar soft-tissue surfaces were slightly higher than those reported here, which was not expected since the present outcomes considered a larger area. As discussed above, higher errors are expected for larger areas compared to smaller areas.

In the present study, a thorough reliability assessment of the entire alveolar soft-tissue surface was performed. Different superimposition reference areas were used for this, covering a wide spectrum of outcomes, with implications for various research and clinical purposes. At first, the reliability of the alveolar process itself, without any consideration of other intraoral structures, was assessed through superimposition on the alveolar surface. This outcome is important when assessments consider only the alveolar structures. Second, alveolar surface reliability was tested following superimpositions on the dental arch. This outcome is of interest when the relations of dental structures with alveolar structures is concerned. For example, the effect of tooth positional changes on the alveolar soft-tissue surface comprises such an outcome. Another assessment concerned the reliability of the alveolar structures following superimposition on the buccal surfaces of the anterior teeth. This area has high accuracy, as previously verified [11], but leads to increased errors, primarily in the palatal areas, as explained earlier. Finally, a reliability assessment of the alveolar surface area was performed following superimposition on adjacent single teeth. The latter approach provided better outcomes when the relation of dental with alveolar structures was considered. Therefore, this should be the technique of choice for relevant outcomes, instead of superimpositions on the entire model or on the entire dental arch. In the latter case, inaccuracies on different areas of the scan are averaged through the overall best fit, skewing the outcomes. Furthermore, the intraoral models show lower accuracy when larger surface areas are considered, as evident in the present study as well as in previous studies [11,13].

Overall, based on the findings of the present study as well as previous studies [18,19], intraoral scans offer a risk-free and highly reliable representation of the actual alveolar surface morphology. Similar findings have been evident for dental [11,17] and palatal structures [6]. Furthermore, if serial intraoral models of an individual are available, techniques have been developed to precisely monitor various outcomes over time, including tooth or material wear [12,13,14,31], tooth positional changes [5,15], gingival width [32], and gingival thickness changes [33,34]. Thus, due to the easy and risk-free acquisition of reliable 3D intraoral models through intraoral scanning and the various uses of this information, we recommend the acquisition of an intraoral model for every patient in early permanent dentition to form the basis for comparisons with future intraoral models. The subsequent data will offer valuable, highly accurate information on dental, palatal, alveolar, and gingival surface changes over time, which can facilitate future research and improve clinical decision making.

This study tested the precision through repeatability and the agreement of two intraoral scanners. Trueness was not assessed because there was no true reference available for comparison with the intraorally obtained surface models. The trueness of the scanners regarding the dentition has been tested against a true reference obtained through high-accuracy industrial scanners and proved to be high [11,17,35]. Therefore, reduced trueness is also not expected in the alveolar surface area, especially when considering the favorable precision and agreement outcomes. A similar trueness assessment was not applicable for gingival surfaces because they could not be captured by the extraoral scanner. A previous in vivo study that assessed the surface of the attached gingiva and the alveolar mucosa at the maxillary buccal anterior area used conventional impressions as a true reference for comparisons [19]. However, the pressure that the impression material exerts on the soft tissues of interest during acquisition can affect the tested surfaces and confound the outcomes [30]. Future research should utilize contactless high-accuracy methods to capture the alveolar soft-tissue morphology and compare it to that obtained through intraoral scanning.

Another limitation of the study is that it tested two widely used scanners from the variety of scanners available in the market. The study also focused on specific measurement areas that were considered to fulfill its purposes. However, the selection of measurement areas affects the generalizability of the outcomes. Other outcomes of interest, such as the gingival margins, might have been selected, but at some point a decision has to be made to keep the number of tests and measurements at a reasonable level. To overcome this limitation, at least partially, all results have been made available in the form of color-coded distance maps.

Finally, to avoid extreme fatigue in the participants, only the maxillary arch was assessed. A similar performance of the scanners for the opposing dental arch might be evident, but this remains to be tested.

## 5. Conclusions

The in vivo reliability of the two intraoral scanners in the alveolar soft-tissue surface area of fully dentate individuals was high, with the difference between repeated scans usually remaining below 50 μm. However, higher errors were observed in a few individual cases, which exceeded 200 μm. The results were only slightly modified when other structures, such as teeth, were used as superimposition references. In this case, single-tooth superimpositions should be preferred for the optimal assessment of neighboring alveolar surface areas relative to the dentition.

The high-quality and risk-free information obtained through intraoral scanning can be used as a reference for comparisons with future models on the alveolar soft-tissue surfaces.

## Figures and Tables

**Figure 1 jcm-11-06389-f001:**
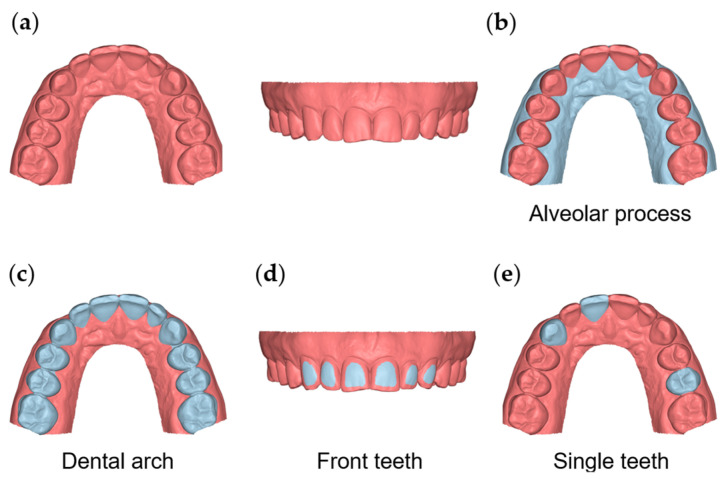
The 3D model and reference areas used in the study. (**a**) The whole 3D model cropped within 7 mm apical to the gingival margin area and distal to the first molar. (**b**) The area of the buccal and palatal gingival and oral mucosa from the gingival margins to 7 mm apically. (**c**) The dental arch area, including all maxillary teeth from first molar to first molar. (**d**) The area of the buccal surfaces of the maxillary anterior teeth. (**e**) Clinical crowns of teeth #11, #13, and #25.

**Figure 2 jcm-11-06389-f002:**
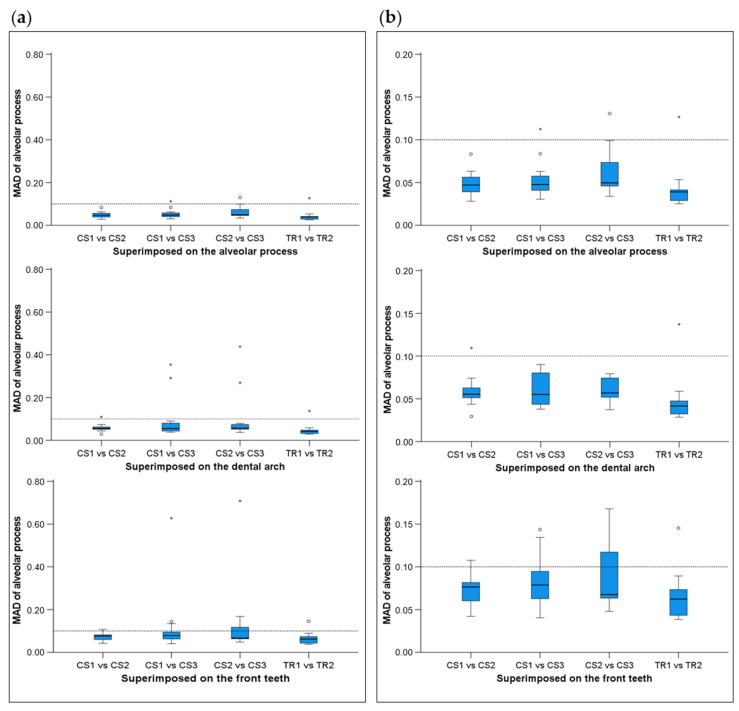
Box plots showing the precision (millimeters) assessment measured through the MAD of the alveolar process between repeated scans with the two different scanners (CS3600 and TRIOS3) when the alveolar process, the whole dental arch, or the front teeth were used as superimposition reference areas. (**a**) Boxplots showing all data. (**b**) Boxplots with magnification, not showing outliers. The upper limit of the black line represents the maximum value, the lower limit represents the minimum value, the box represents the interquartile range, and the horizontal line represents the median value. Outliers are shown as circles or stars in more extreme cases. The dashed horizontal lines indicate 0.1 mm MAD between the compared models. CS1, CS2, and CS3: CS3600 repeated scans. TR1 and TR2: TRIOS3 repeated scans. MAD: Mean Absolute Distance.

**Figure 3 jcm-11-06389-f003:**
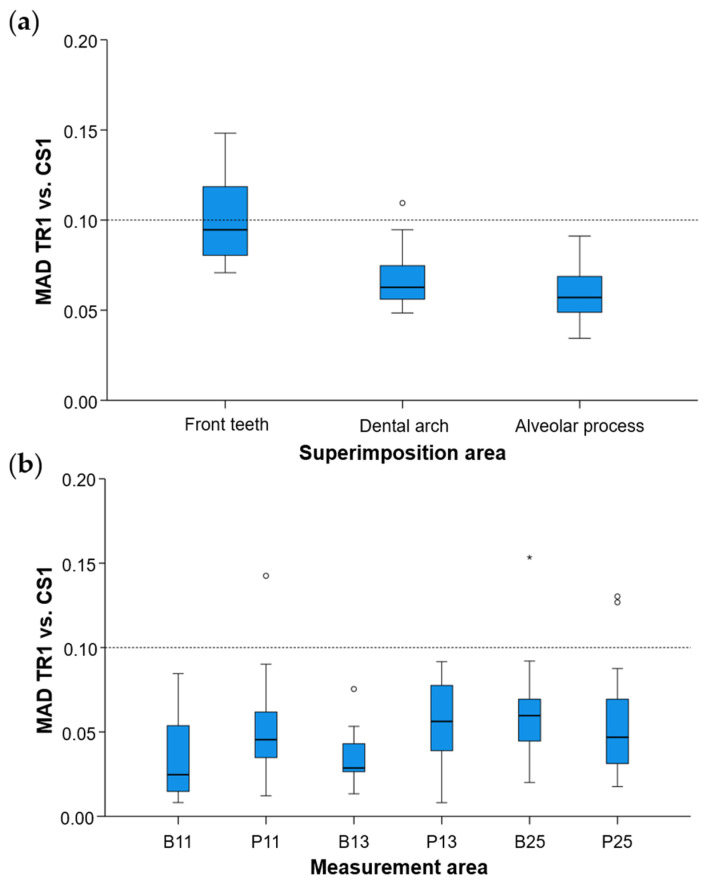
Box plots showing the agreement (millimeters) between the scanners (**a**) measured through the MAD (Mean Absolute Distance) of the alveolar process between repeated scans with the two different scanners (CS1: 1st scanning session with the CS3600 scanner, TR1: 1st scanning session with the TRIOS3 scanner) when the alveolar process, the whole dental arch, or the front teeth were used as superimposition reference areas, and (**b**) at prespecified circular areas located 2 mm apical to the deepest point of the gingival margin area of teeth #11, #13, and #25, at the buccal (B11, B13, B25) and palatal sides (P11, P13, P25). The upper limit of the black line represents the maximum value, the lower limit represents the minimum value, the box represents the interquartile range, and the horizontal line represents the median value. Outliers are shown as circles or stars in more extreme cases.

**Figure 4 jcm-11-06389-f004:**
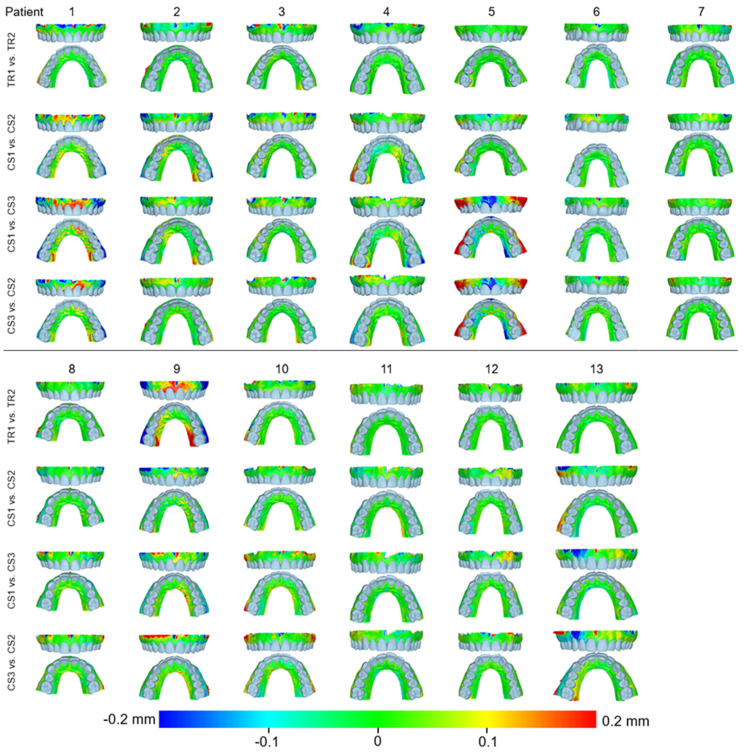
Color-coded distance maps showing the precision of the intraoral scans at the buccal and palatal alveolar soft-tissue surfaces when superimposed on this area. Three scans from CS 3600 (CS1, CS2, and CS3) and two scans from TRIOS3 (TR1 and TR2) were assessed.

**Figure 5 jcm-11-06389-f005:**
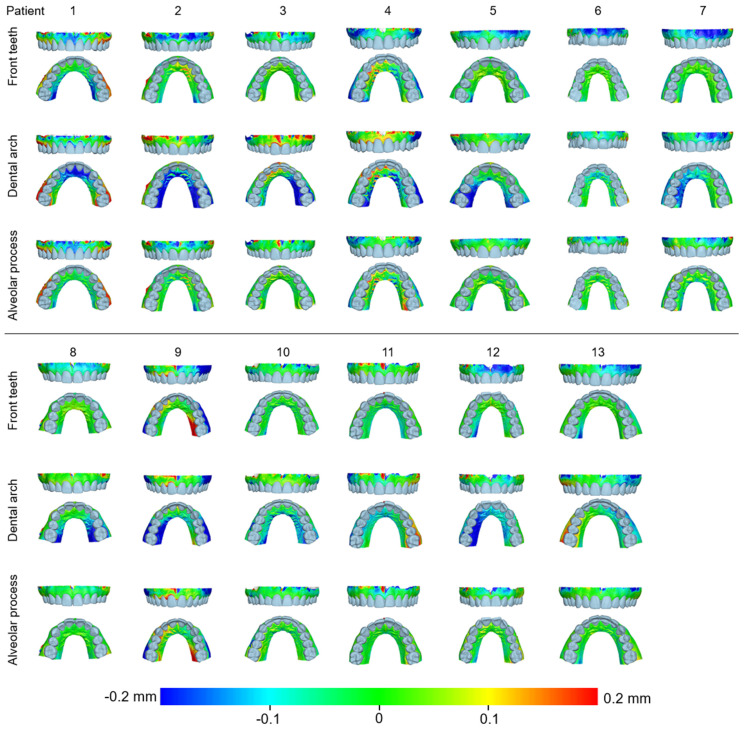
Color-coded distance maps showing the agreement between the intraoral scans at the buccal and palatal alveolar soft-tissue surfaces when superimposed on this area, the front teeth, or the dental arch. Three scans from CS 3600 (CS1, CS2, and CS3) and two scans from TRIOS3 (TR1 and TR2) were assessed.

**Figure 6 jcm-11-06389-f006:**
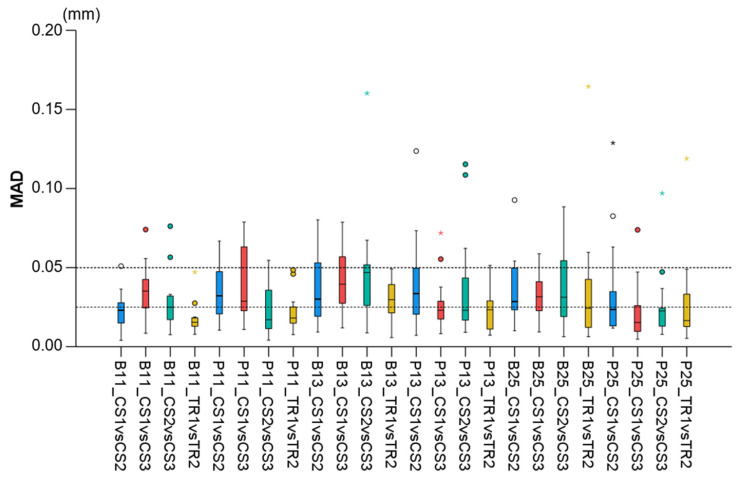
Box plots showing the precision of the intraoral scanners at prespecified circular areas located 2 mm apical to the deepest point of the gingival margin area of teeth #11, #13, and #25 on the buccal (B11, B13, B25) and palatal sides (P11, P13, P25). The upper limit of the black line represents the maximum value, the lower limit represents the minimum value, the box represents the interquartile range, and the horizontal line represents the median value. Outliers are shown as circles or stars in more extreme cases. The dashed horizontal lines indicate 0.025 and 0.05 mm MAD (Mean Absolute Distance) between the compared models.

## Data Availability

All data are available in the main text or the extended data. The protocols and datasets generated and/or analyzed during the current study are available from the corresponding author on reasonable request.

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
