# Peer review of "Intraoral Scanners for In Vivo 3D Imaging of the Gingiva and the Alveolar Process"

_jcm, 2022, doi:10.3390/jcm11216389_

Round 1

Reviewer 1 Report

 This study aimed to assess the precision of two widely used intraoral scanners, as well as their agreement, on the representation of the original morphology of the entire alveolar process in vivo. Complete maxillary scans (CS 3600, Carestream and TRIOS 3, 3Shape) were repeatedly obtained from 13 fully dentate individuals. Scanner precision and agreement were tested using 3D surface superimpositions on the following reference areas: buccal front teeth area, entire dental arch, entire alveolar process, or single teeth, by applying an iterative closest point algorithm. Following each superimposition, the mean absolute distance (MAD) between predefined 3D model surfaces was calculated. Outcomes were analyzed through non-parametric statistics and visualization of color-coded distance maps. When superimpositions were performed on the alveolar process, the median scanner precision was below 0.05 mm, with statistically significant but negligible differences between different scanners. In fact, the agreement between scanners was approximately 0.06 mm. When single teeth superimpositions were used to assess the precision of adjacent alveolar soft/tissue surfaces the overall error was 0.028 mm and there was higher agreement between the scanners. The in vivo reliability of the intraoral scanners in the alveolar surface area was overall high. Single teeth superimpositions should be preferred for optimal assessment of neighboring alveolar surface areas relative to the dentition.

Overall, this study provides insight into whether intraoral scanners can successful scan the entire alveolar process in vivo. As there is an extremely limited number of previous papers on this topic, this new paper is a significant contribution to the existing knowledge.

The research design, methodology and statistical analysis are appropriate for the purpose of this study, the manuscript is overall well written and the illustrations are of high quality.

Reviewer 2 Report

The study design was well-considered and performed, and limitations were apparently described, I think this paper is worthy of processing in the journal if some minor points are clearly improved. And the text also needs English proofreading.

Materials and Methods:

L73, “9M, 4F” is difficult to understand. “9 males and 4 females” is better. 

2.2. Although it is difficult to write, average scan prove speed should be written in this section. Probably, like as “XX-YY cm/sec”.

2.4. You don’t need multiple paragraphs in here, one paragraph is enough. It makes text busy to read.

Results:

Regarding Fig. 2 and 3, labels are quite small and difficult to read. Please enlarge them. They both need the word sizes as Fig.1. Fig. 2 needs bigger images (graphs). 

Fig.2(a), graphs are difficult to read. Decrease the limit value of Y axis and show boxes at the center of the panels. 

Regarding Fig.6, the direction of the labels between X and Y axis is different. Please match them. 

Show significances (ex. *) in all graphs.

Discussion:

It is the era of 3D and computational informatics. In the dentistry, other fields also use the 3D imaging diagnostics. For example, Okuyama et al. reported the importance of making surgical planning of the extraction of ectopic mandibular third molar using both 3D imaging on mandibular bone and inferior alveolar nerve. You can give the short sentences about other usage of 3D technologies citing the article below: 

Okuyama K, Sakamoto Y, Naruse T, Kawakita A, Yanamoto S, Furukawa K, Umeda M. Intraoral extraction of an ectopic mandibular third molar detected in the subcondylar region without a pathological cause: A case report and literature review. Cranio. 2017 Sep;35(5):327-331.

Reviewer 3 Report

This study, entitled “Intraoral scanners for in vivo 3D imaging of the gingiva and the alveolar process”, assessed the precision and agreement of two intraoral scanners (CS 3600 and TRIOS 3) on the soft tissue surfaces of maxillaryalveolar process in 13 fully dentate individualsThe precision and agreement of the two investigated intraoral scannerswere tested after superimposing the maxillary 3D models using three different superimposition methods, respectively. Results showed that the two intraoral scanners achieved high reliability in capturing the alveolar soft tissue surface area of fully dentate individuals with a difference between repeated scans mostly remaining below 50 μm. 

This article is nicely written and structured, very straight forward and easy to understand study-design with a clear discussion and conclusion. There are a few grammatical mistakes should be corrected
